# Mapping health service coverage inequalities in Africa: a scoping review protocol

Humphrey Cyprian Karamagi [ID],[1] Ali Ben Charif,[2] Doris Osei Afriyie,[3] Sokona SY,[1] Hillary Kipruto [ID],[1] Taiwo Oyelade,[1] Benson Droti[1]

[1]WHO Regional Office for Africa, Brazzaville, Republic of Congo
[2]Université Laval, Quebec, Quebec, Canada
[3]Department of Epidemiology and Public Health, Swiss Tropical and Public Health Institute, Allschwil, Switzerland

**Correspondence to**
Dr Humphrey Cyprian Karamagi; karamagih@gmail.com

## ABSTRACT

**Introduction** Addressing inequities in health service coverage is a global priority, especially with the resurgence of interest in universal health coverage. However, in Africa, which has the lowest health service coverage index, there is limited information on the progress of countries in addressing inequalities related to health services. Thus, we seek to map the evidence on inequalities in health service coverage in Africa.

**Methods and analysis** We will conduct a scoping review following the Joanna Briggs Institute Manual for Evidence Synthesis. We preregistered this protocol with the Open Science Framework on 26 July 2022 (https://osf.io/zd5bt). We will consider any empirical research that assesses inequalities in relation to services for reproductive, maternal, newborn and child health (eg, family planning), infectious diseases (eg, tuberculosis treatment) and non-communicable diseases (eg, cervical cancer screening) in Africa. We will search MEDLINE, Embase, Web of Science, CINAHL, PsycINFO and Cochrane Library from their inception onwards. We will also hand-search Google and Global Index Medicus, and screen reference lists of relevant studies. We will evaluate studies for eligibility and extract data from included studies using pre-piloted and standardised forms. We will further extract a core set of health service coverage indicators, which are disaggregated by place of residence, race/ethnicity/culture, occupation, gender, religion, education, socioeconomic status and social capital plus equity stratifiers. We will summarise data using a narrative approach involving thematic syntheses and descriptive statistics. We will report our findings according to the Preferred Reporting Items for Systematic Reviews and Meta-Analyses extension for Scoping Reviews checklist.

**Ethics and dissemination** Ethical approval is not required as primary data will not be collected. This work will contribute to identifying knowledge gaps in the evidence of inequalities in health service coverage in Africa, and propose strategies that could help overcome current challenges. We will disseminate our findings to knowledge users through a publication in a peer-reviewed journal and organisation of workshops.

## STRENGTHS AND LIMITATIONS OF THIS STUDY

⇒ This scoping review will follow the Joanna Briggs Institute Manual for Evidence Synthesis.
⇒ We will conduct a comprehensive literature on multiple electronic databases on inequalities and service coverage in Africa.
⇒ We will use place of residence, race/ethnicity/culture, occupation, gender, religion, education, socioeconomic status and social capital plus to guide our search strategy and conceptualisation of inequality.
⇒ We will adapt the service coverage indicators used in tracking universal health coverage by the WHO and World Bank.
⇒ We foresee extensive data, given the broad indicators for service coverage.

## INTRODUCTION

The attainment of good health and well-being has been prioritised as a common goal by African countries, as outlined in the third Sustainable Development Goal established by the United Nations in 2015.[1] Within the WHO Regional Office for Africa, countries have recognised attainment of universal health coverage (UHC) as a critical outcome necessary to attain this goal along with good health security and coverage of health determinants.[2 3] One of the main goals of UHC is to ensure that all people receive the health services they need, including promotive, preventative, curative, rehabilitative and palliative care, which are of sufficient quality.[4] At the core of UHC goals is a commitment to health equity.

WHO defines health equity as 'the absence of unfair and avoidable or remediable differences in health among population groups defined socially, economically, demographically or geographically'.[5] In principle, health inequities are systematic differences that are socially produced, and put groups disadvantaged already at further disadvantage related to their health.[6] A key step toward addressing and assessing health equities is monitoring health inequalities–health differences between population subgroups.[7]

The roots of inequalities in health can be complex and influenced by a myriad of social

conditions. In 2005, the WHO commission on social determinants of health emphasised the role of structural mechanisms, which create stratification and social class divisions that shape the health opportunities of various social groups based on their level of power, prestige and access to resources.[5] The commission identified six important structural stratifiers: (1) income, (2) education, (3) occupation, (4) social class, (5) gender and (6) race or ethnicity. Additionally, other studies in sub-Saharan Africa have also recognised the need to include historical and cultural context, which underlies causal factors for the social determinants of health in the region.[8 9]

Few reviews have specifically examined inequalities within the context of UHC. The limited reviews that have been conducted have mainly focused on specific services such as reproductive, maternal, newborn and child health (RMNCH) services.[10 11] Additionally, others have assessed inequalities using selected stratifiers, such as socioeconomic status and age.[12 13] In this review, we seek to consolidate the evidence on service coverage inequalities in Africa using a comprehensive set of stratifiers to assess these inequalities. The specific objectives of the review are to: (1) outline the methodological approaches used in assessing health inequalities in relation to service coverage; (2) characterise the current evidence on service coverage inequalities; (3) identify knowledge gaps in the existing evidence on service coverage inequalities); (4) document effective strategies being used to tackling the different drivers of inequalities in service coverage and (5) identify challenges related to addressing health equalities in Africa.

## METHODS AND ANALYSIS
### Design
We will conduct a scoping review following the methodology recommended in the Joanna Briggs Institute (JBI) Manual for Evidence Synthesis.[14] This methodology is based on the Arksey and O'Malley's framework[15] and an enhanced version developed by Levac and colleagues[16] for conducting scoping reviews.[17] A scoping was selected as we aim to outline the evidence in the area of interest and identify knowledge gaps. This protocol has been registered with the Open Science Framework (OSF) on 26 July 2022 (identifier: https://osf.io/zd5bt). We searched our electronic databases on 29 August 2022, and plan to complete this review by 26 June 2023. We will report this review according to the Preferred Reporting Items for Systematic Reviews and Meta-Analyses (PRISMA) extension for Scoping Reviews checklist.[18] In this protocol, we use the standard PRISMA definitions for a report.[19] We report the content for this scoping review protocol using the PRISMA Protocol checklist (online supplemental appendix 1).[20]

### Eligibility criteria
Following the JBI Manual for Evidence Synthesis,[14] we will use the following eligibility criteria:

► Participants: we will consider studies involving individuals, communities or organisations involved in the receipt of health services within a health system context in Africa. No restrictions based on sociodemographic factors (eg, sex, age and ethnicity) or health conditions (eg, comorbidities) will be applied. We will consider any countries or geographic regions in the African continent such as the 47 member states of the WHO Regional Office for Africa, the Maghreb and all other African regions. Global studies which include both an African region and other regions from other continents (eg, Europe, Asia or America) will be considered. We will exclude studies involving only regions or countries outside the African geographic region.

► Context: we will consider studies that assess the use of essential health services. We will adapt the WHO and World Bank indicators for health service coverage and will consider essential health services within three components: (1) RMNCH (family planning, antenatal care, delivery care, postnatal care, child immunisation and health-seeking behaviour for pneumonia); (2) infectious diseases (tuberculosis treatment, HIV therapy, use of insecticide-treated bed nets for malaria prevention, adequate sanitation and neglected tropical diseases treatment) and (3) non-communicable diseases (prevention and treatment of raised blood pressure, prevention and treatment of raised blood glucose, cervical cancer screening and tobacco (non-) smoking) (online supplemental appendix 2). We will also consider any index or sub-index assessing health service coverage of these components (eg, sub index on infectious disease services). However, we will exclude indicators or indices related to basic hospital access, health worker density, access to essential medicines or health security.

► Types of sources: we will consider empirical studies using qualitative, quantitative or mixed methods designs. This includes knowledge syntheses (eg, systematic reviews), experimental, quasi-experimental and observational designs. No restrictions will be placed on the language of publication or publication status. However, we will exclude studies published before 2005, because that is the year the term 'Universal Coverage' was mentioned in a World Health Assembly resolution.[21] Additionally, we will also exclude any retracted publications, conference abstracts, study protocols and editorial materials (eg, editorials, commentaries and letters).

► Concept: we will consider studies assessing inequalities or differences in health service coverage between subgroups. Health service coverage refers to the access to or use of health services (ie, equal service for equal need).[22 23] To identify stratifiers, we will adopt the Cochrane and Campbell Equity Methods group framework of place of residence, race/ethnicity/culture, occupation, gender, religion, education, socioeconomic status and social capital (PROGRESS)

**Table 1** Criteria for considering studies for this review

| Criteria | Inclusion | Exclusion |
|---|---|---|
| Participants | Individuals, communities or organisations involved in the receipt of healthcare services within a health system context in Africa | ▶ Any countries or regions outside the African geographic region |
| Concept | Studies focusing on one of the following PROGRESS-Plus* equity stratifier:<br>▶ Place of residence (eg, rural and urban)<br>▶ Race, ethnicity or culture<br>▶ Occupation<br>▶ Gender or sex<br>▶ Religion<br>▶ Education<br>▶ Socioeconomic status<br>▶ Social capital or resources<br>▶ Any other factors in which health inequalities may exist (eg, age, disability and sexual orientation). | ▶ Studies that did not include a PROGRESS-Plus* equity stratifier<br>▶ Studies that did not examine inequalities in health service coverage<br>▶ Studies focusing on inequality in health financing or financial protection |
| Context | Studies monitoring at least one indicator related to the following essential health services (online supplemental appendix 2):<br>▶ Family planning<br>▶ Antenatal, delivery and postnatal care<br>▶ Child immunisation<br>▶ Health-seeking behaviour for pneumonia<br>▶ Tuberculosis treatment<br>▶ HIV therapy<br>▶ Use of insecticide-treated bed nets for malaria prevention<br>▶ Adequate sanitation<br>▶ Neglected tropical diseases treatment and care<br>▶ Prevention and treatment of raised blood pressure<br>▶ Prevention and treatment of raised blood glucose<br>▶ Cervical cancer screening<br>▶ Tobacco (non-)smoking | ▶ Studies that did not include any of those essential health services<br>▶ Studies focusing on basic hospital access<br>▶ Studies focusing on health workforce<br>▶ Studies focusing on access to essential medicines<br>▶ Studies focusing on health security |
| Type of sources | Empirical studies using a quantitative, qualitative or mixed methods design and published from 2005:<br>▶ Original studies from 2005 onwards<br>▶ Conference articles from 2005 onwards<br>▶ Knowledge syntheses from 2005 onwards | ▶ Protocols<br>▶ Conference abstracts<br>▶ Retracted publications<br>▶ Records published before 2005<br>▶ Editorial materials (eg, commentary, letter and editorials) |

*The 'plus' includes other factors in which health inequalities may exist such as age, disability and sexual orientation.
PROGRESS, place of residence, race/ethnicity/culture, occupation, gender, religion, education, socioeconomic status and social capital.

plus. This framework includes social factors that can influence health disparities and the 'plus' includes other factors in which health inequalities may exist such as age, disability and sexual orientation. We will exclude studies that did not examine inequalities in health service coverage, such as studies on health financing or financial protection, an area overviewed in the literature. However, studies which assess financial hardship as a driver of inequalities of health service will be included.

In essence, we will consider any empirical research that uses any study designs and measures inequalities in relation to services for services for RMNCH (eg, family planning), infectious diseases (eg, tuberculosis treatment) and non-communicable diseases (eg, cervical cancer screening) in Africa (table 1).

**Literature search**
We will perform a comprehensive search to identify records through electronic databases and other relevant sources. No restrictions will be placed on date of publication, language, place of publication or type of reports in our search strategy.

We will search MEDLINE, Embase, Web of Science, CINAHL, PsycINFO and the Cochrane Library from their dates of inception onwards. We will perform the preliminary search strategy in Ovid MEDLINE following appropriate design principles.[24] An information specialist and our core team of international experts in health equity, UHC, health information systems or knowledge syntheses from Africa will review this preliminary search strategy. The search terms will be adapted to the above-mentioned databases. The search terms will be based on previous

works to reflect three concepts: (1) inequality, (2) health service coverage and (3) African regions (online supplemental appendix 3). For inequality, we will adapt a validated search filter,[25] by including adding other terms that are suitable for the African context (eg, rural and religion).[26–28] For service coverage, we will use terms related to UHC that were previously identified with an exploratory search in Google, Google Scholar and Abstract reviews.[29] For African geographic regions, we will adapt a geographic African filter validated by the South African Cochrane Centre,[30] by correcting the name 'Mayotte' and including alternative missing African country names (eg, 'Ruanda', 'Comoros' and 'Cabo Verde'). We will use the list of African region names used to develop the low-income and middle-income countries' geographic search filter by the Cochrane Effective Practice and Organisation of Care, in collaboration with the WHO and Campbell Collaboration.[31 32]

In addition to electronic databases, we will also identify relevant records through screening reference lists of relevant reports and hand searching on Google and WHO Global Index Medicus. From the results of the two websites, we will screen at least the first 30 results for each search. Previous experiences show that results beyond the first 30 results are often duplicates and unlikely to be relevant.[33 34]

### Selection of sources of evidence
Following of the search, we will collate and upload all the records all the records into EndNote V.20 (Clarivate Analytics, PA, USA), and remove duplicates. Screening forms, standardised in Google Sheets, will be prepared based on eligibility criteria refined by the entire review team to ensure accurate selection of eligible records. As suggested by JBI, we will select a random sample of 25 records for the pilot test and only start screening when agreement of 75% or greater is achieved. We will calculate the inter-reviewer agreement using the weighted Cohen's kappa.[35] One reviewer will screen all remaining records and identify potentially relevant reports that meet the eligibility criteria. Each record or report will be screened by one using the standardised forms and checked by another. We will document a reason for excluding any ineligible report. Any discrepancies will be resolved through consensus or with the assistance of a third reviewer.

### Data charting process
We will develop a form in Google sheets in consultation with the core team to guide the extraction of variables. Two reviewers will independently perform a pilot test of the form to ensure it captures relevant data. We will extract the following information: study characteristics (eg, title, authors, year of publication, design, target participants and country); inequality dimensions (eg, PROGRESS-Plus elements); methodological approaches used to measure inequalities (eg, indices) and health service coverage indicators (eg, skilled birth attendance and complete antenatal care visits). Full charting will be completed by one reviewer and checked by another. Any discrepancies between reviewers will be resolved by discussion or with the assistance of a third reviewer.

### Critical appraisal
Due to the nature of our research question, we will not perform an appraisal for risk of bias or conduct quality assessment. This is consistent with the JBI Manual for Evidence Synthesis.[14] A critical appraisal is generally not recommended in scoping reviews because the aim is to map the available evidence rather than provide a synthesised and clinically meaningful answer to our research question.[36]

### Synthesis of results
We will employ both qualitative and quantitative methods to analyse the data generated. A descriptive summary of the characteristics of included studies will be presented.[16] We will map studies according to the appropriate health service indicators and PROGRESS-Plus elements. Additionally, we will undertake a qualitative synthesis to identify common themes among included studies on the evidence in the findings and probable explanations for service coverage inequalities in the discussion sections.[16] We will use the PRISMA 2020 flowchart to describe the process of report selection.[19]

### Patient and public involvement
Patients and the public will not be involved in the design, conduct or parting of this scoping review. However, we will adopt an integrated knowledge translation approach, where policy-makers, clinicians, researchers and trainees are equal members. This will ensure that the research is relevant and useful to knowledge users, increasing the likelihood of uptake.[37] This will involve female and male international African experts. They will be engaging virtually once a week to discuss status and progress that will be transparently available to all members using google drive. Multidisciplinary consultations will also be conducted with policy-level experts who will be purposively selected based on the topic for analysis, to enrich the interpretation of findings.

### DISCUSSION
This scoping review seeks to map the evidence on the state of inequality in the progress toward universal health service coverage and its constituent components in Africa. It will fill an important gap by providing a comprehensive body of evidence that exists on the progress toward equity in UHC across Africa. First, it will be possible to map the evidence on inequalities in coverage for the broad range of essential services across several dimensions of inequalities. Second, it will be able to highlight the barriers and opportunities for effectively addressing the drivers of health inequalities in Africa. Third, it will be possible to demonstrate the appropriate methodological approaches

for measuring health inequalities in the African context. Lastly, it will highlight which essential services and its relevant dimensions of inequalities that have been under-researched in the literature, and may need future investigations. Thus, this review will make a critical contribution to monitoring inequalities in health service coverage promoting learning and building the evidence for investments in effective strategies to reduce health disparities in Africa.

## Ethics and dissemination

This scoping review will involve neither human participants nor unpublished secondary data. As such, formal ethical approval from a research ethics committee is not required. We will disseminate our results through publications in peer-reviewed journals and a technical report for the WHO Regional Office for Africa. We will also share our reports using free public repositories such as OSF and ResearchGate.

**Acknowledgements** We wish to acknowledge the WHO Regional Office for Africa for their assistance with various aspects of this work.

**Contributors** HCK, SS, HK, TO and BD conceptualised the study. HCK, ABC, DOA, SS and HK participated in the design of this protocol. ABC drafted the search strategy and all authors revised it. ABC and DOA drafted this protocol. All authors revised the manuscript critically for important intellectual content, gave final approval of the version to be published and agreed to be accountable for all aspects of the knowledge synthesis.

**Funding** The authors have not declared a specific grant for this research from any funding agency in the public, commercial or not-for-profit sectors.

**Disclaimer** The author is a staff member of the World Health Organization. The author alone is responsible for the views expressed in this publication and they do not necessarily represent the views, decisions or policies of the World Health Organization.

**Competing interests** None declared.

**Patient and public involvement** Patients and/or the public were not involved in the design, or conduct, or reporting, or dissemination plans of this research.

**Patient consent for publication** Not applicable.

**Provenance and peer review** Not commissioned; externally peer reviewed.

**ORCID iDs**
Humphrey Cyprian Karamagi http://orcid.org/0000-0002-6277-2095
Hillary Kipruto http://orcid.org/0000-0002-3879-1712

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
