## [Reviewer comments · BMJ Open]

ARTICLE DETAILS

TITLE (PROVISIONAL)	Mapping health service coverage inequalities in Africa: a scoping review protocol
AUTHORS	Karamagi, Humphrey Cyprian; Ben Charif, Ali; Afriyie, Doris; SY, Sokona; kipruto, Hillary; Oyelade, Taiwo; Droti, Benson

VERSION 1 – REVIEW

REVIEWER	Ayiasi, Mangwi Richard Inst Trop Med
REVIEW RETURNED	04-Nov-2022

GENERAL COMMENTS	Line 90 defining Health inequities as the "absence of... unavoidable remedial..." is quite confusing. The word "unavoidable" looks misplaced! Line 100 -101: The authors introduce multiple concepts "...either focused on specific services, or drivers of inequality. It would simplify issues by mentioning some of these "services" and some of the "drivers of inequality" Line 103 to 116 provides a narration of conceptual thoughts. The authors should consider mapping this out in pictorial/diagram for easier comprehension for the reader Line 181-182: Justify reason for exclusion of equity in health financing and financial exclusion, yet this seems to be the essence of this study - financial protection! Line 212 does not read well. Please check
--

REVIEWER	Pratiwi, Agnes Universitas Gadjah Mada Fakultas Kedokteran, Medical Education and Bioethics
REVIEW RETURNED	15-Jan-2023

GENERAL COMMENTS	Thank you for the opportunity to review this important study protocol. I enjoyed reading how the authors clearly articulate the gap in the research in this area and the details of the methods. The study is extensive and would be relevant to the region and beyond. However, a substantive note in the inconsistent or interchangeable use of terms between inequity, inequality, and equity will make the readers difficult to understand the results later, or difficulties of other researchers who want to replicate a similar research in future. I would like to make few comments: -How will you treat the different credibility of evidence, for example, peer-reviewed articles vs reports vs articles published in non-reputable journals? Please also explain clearly why critical appraisal or a quality check on the articles will not be performed.
--

	-Although the concepts differ, equity, inequity, and inequality are used mixed or interchangeably. Meanwhile, in articles, it may have different meanings and thus methods in measurements. -Line 100. "At present, most evidence is either focused on specific services, or drivers of inequality." – please support with references. -Line 128-136. Objective one is about methodological approaches on inequality, meanwhile objectives two and three are about "equity". -How will each objective technically translate into a do-able search strategy and thus yield the related evidence? -Line 133. "...and propose strategies that could help overcome current challenges." – This might fit better to a separate objective? -Line 159 "whose health needs are supposed to be addressed" – what are the criteria? -Line 181. "We will exclude studies focusing on equity in health financing or financial protection." As financial protection is one dimension of Universal Health Coverage, could you explain why this is excluded? -In the method section, search strategy used "equity" but the objectives are also about inequality. -Hand searching involves manually opening the resources page per page to seek articles, for example, a journal edition. Will this be performed? -Objective 1 stated mapping methods in measuring health inequalities, but in the charting "methodological approach used to measure equality (e.g., indices)." - This is different. -"the study findings and discussions will be analyzed using content analysis to develop codes and themes that emerge from the data." – could you describe how findings and discussions will be treated, for which particular purpose you will analyze the findings and for which the discussion section?
--	--

VERSION 1 – AUTHOR RESPONSE

Reviewer 1's Comments (Dr. Mangwi Richard Ayiasi, Inst Trop Med)	How they are addressed
Line 90 defining Health inequities as the "absence of... unavoidable remedial..." is quite confusing. The word "unavoidable" looks misplaced!	Thank you for identifying this error. We have corrected the word as follows (Page 4): “WHO defines health equity as “the absence of unfair and unavoidable or remediable differences in health among population groups defined socially, economically, demographically or geographically” [5].”
Line 100 -101: The authors introduce multiple concepts "...either focused on specific services, or drivers of inequality. It would simplify issues by mentioning some of these "services" and some of the "drivers of inequality"	Thank you for highlighting this. We have removed this sentence as we had a sentence on line 122 that was more explicit. We have also revised this sentence as follows (Page 5): “The limited reviews that have been conducted have mainly focused on specific services such as Reproductive, Maternal, Newborn and Child health (RMNCH) services [10,11]. Additionally, others have assessed inequalities using selected stratifiers, such as socioeconomic status and age [12,13].”
Line 103 to 116 provides a narration of conceptual thoughts. The authors should consider mapping this out in pictorial/diagram for easier comprehension for the reader	Thank you for this comment. We agree that this paragraph may be difficult for readers to comprehend. Therefore, we have revised it to be clearer and easier to understand. This paragraphs now reads as follows (Pages 4): “The roots of inequalities in health can be complex and influenced by a myriad of social conditions. In 2005, the WHO commission on social determinants of health emphasized the role of structural mechanisms, which create stratification and social class divisions that shape the health opportunities of various social groups based on their level of power, prestige and access to resources [5]. The commission identified six important structural stratifiers: 1) income, 2) education, 3) occupation, 4) social class, 5) gender, and 6) race or ethnicity. Additionally, other studies in sub-Saharan Africa have also recognized the need to include historical and cultural context, which underlies causal factors for the social determinants of health in the region [8,9].”

Line 181-182: Justify reason for exclusion of equity in health financing and financial exclusion, yet this seems to be the essence of this study - financial protection!	Thank you for raising this important point, also highlighted by Reviewer 2. However, previous knowledge syntheses have prioritized financial protection (Bhatia et al. 2022; http://dx.doi.org/10.1136/bmjopen-2021-052041). Furthermore, within the African region, health service coverage is one of the lowest. For this reason, the WHO Regional Office for Africa suggested focusing this review on use of health services. First, as suggested, we have added a justification in the Methods section (Page 7): “We will exclude studies that did not examine inequalities in health service coverage, such as studies on health financing or financial protection, an area overviewed in the literature.” Second, in the Introduction section, we have made it clearer that we will be focusing on service coverage (Page 4): “The attainment of good health and well-being has been prioritized as a common goal by African countries, as outlined in the third Sustainable Development Goal (SDG 3) established by the United Nations in 2015 [1]. Within the World Health Organization (WHO) Regional Office for Africa, countries have recognized attainment of universal health coverage (UHC) as a critical outcome necessary to attain this goal along with good health security and coverage of health determinants [2,3]. One of the main goals of universal health coverage is to ensure that all people receive the health services they need, including promotive, preventative, curative, rehabilitative, and palliative care which are of sufficient quality [4]. At the core of universal health coverage goals is a commitment to health equity.” Finally, we will be assessing an extensive list of health service coverage indicators (see Appendix 2) and including financial protection will be too broad of a research question For example, to this day, we have identified 188 eligible reports.
---	---

Line 212 does not read well. Please check.	Thanks for this comment. We have rephrased both sentences as follows (Page 10): “...we will exclude studies published before 2005, because that is the year the term “Universal Coverage” was mentioned in a World Health Assembly resolution [23]. Additionally, we will also exclude any retracted publications, conference abstracts, study protocols, and editorial materials (e.g., editorials, commentaries, and letters).”
Reviewer 2's Comments (Dr. Agnes Pratiwi, Universitas Gadjah Mada Fakultas Kedokteran, Academisch Medisch Centrum)	How they are addressed
Thank you for the opportunity to review this important study protocol. I enjoyed reading how the authors clearly articulate the gap in the research in this area and the details of the methods. The study is extensive and would be relevant to the region and beyond. However, a substantive note in the inconsistent or interchangeable use of terms between inequity, inequality, and equity will make the readers difficult to understand the results later, or difficulties of other researchers who want to replicate a similar research in future. I would like to make few comments:	Thank you very much for your suggestions that have helped us to propose a revised and improved version of the manuscript. As suggested, we have reviewed the entire manuscript in order to correct the inconsistent use of terminologies and facilitate reading and understanding. In the Introduction section, we have described that health equity is the main goal of universal health coverage, but in order to address or assess it, we need to measure health inequalities. We have made it clearer in the protocol that we will be assessing health inequalities in relation to service coverage.
How will you treat the different credibility of evidence, for example, peer-reviewed	Thank you for this comment. A critical appraisal is generally not recommended in scoping reviews because the aim is to map the available evidence rather than provide a synthesized and clinically

articles vs reports vs articles published in non-reputable journals?	meaningful answer to a question (Peters et al. JBI Evidence Synthesis, 2020; Peters et al. JBI Manual for Evidence Synthesis, 2020; Peters et al. JBI Reviewer's Manual, 2015, 2017; Khalil et al. Worldviews Evid Based Nurs, 2016). For this reason, an assessment of methodological limitations of the evidence included within a scoping review is generally not performed.
Please also explain clearly why critical appraisal or a quality check on the articles will not be performed.	As suggested, we have added those clarifications as follows (Page 12): “Due to the nature of our research question, we will not perform an appraisal for risk of bias or conduct quality assessment. This is consistent with the Joanna Briggs Institute (JBI) Manual for Evidence Synthesis [14]. A critical appraisal is generally not recommended in scoping reviews because the aim is to map the available evidence rather than provide a synthesized and clinically meaningful answer to our research question [36].”
Although the concepts differ, equity, inequity, and inequality are used mixed or interchangeably. Meanwhile, in articles, it may have different meanings and thus methods in measurements.	Thank you for pointing this out. First, we agree that these concepts may be used interchangeably. Thus, we have reviewed the entire manuscript in order to correct the inconsistent use of terminologies and facilitate reading and understanding. Second, in our method section, we have stated that we will consider studies assessing inequalities or differences in health service coverage between subgroups. Our focus will be on the measurement of group differences and not on terminology used. We will outline the methodologies used in measurements. Finally, in our search terms, “equity” represented a broad concept including terms such as equity, inequity, equality, inequality, disparity, and deprivation (see Appendix 3).
Line 100. “At present, most evidence is either focused on specific services, or	Thank you for highlighting this. We have removed this sentence as we had a sentence on line 122

drivers of inequality.” – please support with references.	that was more explicit. We have also revised this sentence to (Page 5): “The limited reviews that have been conducted have mainly focused on specific services such as Reproductive, Maternal, Newborn and Child health (RMNCH) services [10,11]. Additionally, others have assessed inequalities using selected stratifiers, such as socioeconomic status and age [12,13].”
Line 128-136. Objective one is about methodological approaches on inequality, meanwhile objectives two and three are about “equity”. How will each objective technically translate into a do-able search strategy and thus yield the related evidence?	Thank you for raising this point. We have revised the entire manuscript to be consistent and clear that we are interested in assessing health inequalities in regards to health service coverage. The objectives now read as follows (Page 5): “In this review, we seek to consolidate the evidence on service coverage inequalities in Africa using a comprehensive set of stratifiers to assess these inequalities. The specific objectives of the review are to: 1) Outline the methodological approaches used in assessing health inequalities in relation to service coverage; 2) Characterize the current evidence on service coverage inequalities; 3) Identify knowledge gaps in the existing evidence on service coverage inequalities; 4) Document effective strategies being used to tackling the different drivers of inequalities in service coverage; and 5) Identify challenges related to addressing health equalities in Africa.”
Line 133. “...and propose strategies that could help overcome current challenges.” – This might fit better to a separate objective?	Thank you for the suggestion. We have made it a separate objective.
Line 159 “whose health needs are supposed to be addressed” – what are the criteria?	Thank you for pointing this out. We have rephrased this sentence because we will not use a criteria to assess whether health needs are supposed to be addressed or not. This sentence now reads as follows (Page 6): “We will consider studies involving individuals, communities, or organizations involved in the receipt of health services within a health system context in Africa.”

Line 181. “We will exclude studies focusing on equity in health financing or financial protection.” As financial protection is one dimension of Universal Health Coverage, could you explain why this is excluded?	Thank you for raising this important point, also highlighted by Reviewer 1. We strongly agree that equity in financial protection is an essential component of UHC that needs to be assessed. However, to the best of our knowledge, published knowledge syntheses overviews of reviews (reviews of reviews) have prioritized financial protection over service coverage (Bhatia et al. 2022; http://dx.doi.org/10.1136/bmjopen-2021-052041). For this reason, the WHO Regional Office for Africa suggested focusing this review on use of health services, an area of unexplored potential. First, as suggested, we have added a justification in the Methods section (Page 7): “We will exclude studies that did not examine inequalities in health service coverage, such as studies on health financing or financial protection, an area overviewed in the literature.” Second, in the Introduction section, we have made it clearer that we will be focusing on service coverage (Page 4): “The attainment of good health and well-being has been prioritized as a common goal by African countries, as outlined in the third Sustainable Development Goal (SDG 3) established by the United Nations in 2015 [1]. Within the World Health Organization (WHO) Regional Office for Africa, countries have recognized attainment of universal health coverage (UHC) as a critical outcome necessary to attain this goal along with good health security and coverage of health determinants [2,3]. One of the main goals of universal health coverage is to ensure that all people receive the health services they need, including promotive, preventative, curative, rehabilitative, and palliative care which are of sufficient quality [4]. At the core of universal health coverage goals is a commitment to health equity.” Finally, we will be assessing an extensive list of health service coverage indicators (see Appendix 2) and including financial protection will be too broad of a research question For example, to this day, we have identified 188 eligible reports.
---	---

In the method section, search strategy used “equity” but the objectives are also about inequality.	Thank you for pointing this out. We agree and have revised the entire manuscript to be consistent and clear that we are interested in assessing health inequalities in regards to health service coverage. The objectives now read as follows (Page 5): “In this review, we seek to consolidate the evidence on service coverage inequalities in Africa using a comprehensive set of stratifiers to assess these inequalities. The specific objectives of the review are to: 1) Outline the methodological approaches used in assessing health inequalities in relation to service coverage; 2) Characterize the current evidence on service coverage inequalities; 3) Identify knowledge gaps in the existing evidence on service coverage inequalities; 4) Document effective strategies being used to tackling the different drivers of inequalities in service coverage; and 5) Identify challenges related to addressing health equalities in Africa.” Additionally, our search terms reflect three concepts: 1) equity, 2) universal health coverage, and 3) African regions (see Appendix 3). As we understand studies often mix or use the concepts interchangeably, we made our search strategy comprehensive by including all these concepts or terms such as inequity, equity, health disparities, deprivation and equality.
Hand searching involves manually opening the resources page per page to seek articles, for example, a journal edition. Will this be performed?	Thank you for this comment. We meant hand searching for records or reports on Google and WHO Global Index Medicus. Indeed, we have attempted to expand our review to some sources, including publicly available information produced by all levels of government, academic institutions, business, and industry, in print and electronic formats, which are not controlled by commercial publishers. We have made the sentence clearer as follows (Page 11): “In addition to electronic databases, we will also identify relevant records through screening reference lists of relevant reports and hand searching on Google and WHO Global Index Medicus. From the results of the two websites, we will screen at least the first 30 results for each search. Previous experiences show that results beyond the first 30 results are often duplicates and unlikely

	to be relevant [33,34].”
Objective 1 stated mapping methods in measuring health inequalities, but in the charting “methodological approach used to measure equality (e.g., indices).” - This is different.	Thank you for identifying this error. We have made both sentences consistent by revising to (Page 5): “...Outline the methodological approaches used in assessing health inequalities in relation to service coverage;...”
“the study findings and discussions will be analyzed using content analysis to develop codes and themes that emerge from the data.” – could you describe how findings and discussions will be treated, for which particular purpose you will analyze the findings and for which the discussion section?	Thank you for this comment. The content analysis is a component of the qualitative synthesis of the findings. Its purpose is to identify common themes among the included studies in relation to their findings on service coverage inequalities. For the discussion section, the content analysis will help us to identify common probable explanations offered by the authors for their findings. We have revised this sentence as follows (Page 13): “...we will undertake a qualitative synthesis to identify common themes among included studies on the evidence in the findings and probable explanations for service coverage inequalities in the discussion sections [16].”

VERSION 2 – REVIEW

REVIEWER	Pratiwi, Agnes Universitas Gadjah Mada Fakultas Kedokteran, Medical Education and Bioethics
REVIEW RETURNED	24-Apr-2023

GENERAL COMMENTS	Dear Authors, Thank you for the revisions. The manuscript is now presented accurately and in enough detail for general readers and researchers who wish to understand the details. It is now possible to understand quickly that the focus is on health inequalities, and the authors have clearly explained the different terms around it conceptually and technically. It is also a strength of this research that the authors focused primarily on service coverage instead of financial protection. I recommend this version for publication. I look forward to reading the published version of the protocol and the results of this critical review on health inequalities in Africa.
---